# Understanding Medication Errors in Intensive Care Settings and Operating Rooms—A Systematic Review

**DOI:** 10.3390/medicina61030369

**Published:** 2025-02-20

**Authors:** Katarzyna Kwiecień-Jaguś, Wioletta Mędrzycka-Dąbrowska, Monika Kopeć

**Affiliations:** 1Department of Anaesthesiology Nursing & Intensive Care, Faculty of Health Sciences, Medical University of Gdansk, 80-211 Gdansk, Poland; 2Department of Human Nutrition, University Warmia and Mazury, 10-718 Olsztyn, Poland

**Keywords:** medical staff, medication error, intensive care unit (ICU), operating room (OR), patient safety

## Abstract

*Background and Objectives:* A medication error can occur at any stage of medication administration at the ward, from the moment the medication is prescribed through the preparation to the administration to the patient. The statistics indicate that the scale of the problem, which has a significant impact on the safety and health of patients, is still poorly known. The purpose of the systematic review was to synthesise the published research about the number of medication errors in operating room theatres and intensive care units. *Materials and Methods:* The literature review was conducted in the third quarter of 2023. The overview included papers found in Science Direct, EBSCO, PubMed, Ovid, Scopus, and original research papers published in English meeting the PICOS criteria. Original articles published between 2017 and 2023 that meet the inclusion criteria were included for further analysis. *Results:* The review included 13 articles and original studies, which met the PICOS-based criteria. The analyses confirmed that the operating theatre’s medication error rate was 7.3% to 12%. In the case of intensive care units, the medication error rate was from 1.32 to 31.7%. *Conclusions:* Medication errors in the operating room and intensive care are high. However, the values presented herein do not differ from the general Medication Error Index for medical centres, as calculated by the World Health Organization.

## 1. Introduction

Medication errors (MEs) are one of the basic issues of health care in the world and have a real impact on patient safety [1]. The definition of a medication error was used by the US National Coordinating Council for Medication Error Reporting and Prevention in 2002. A medication error is defined as any preventable event that may cause or lead to inappropriate medication use or patient harm while the medication is in the control of the healthcare professional, patient, or consumer [2]. A medication error can occur at each stage of medicine administration, i.e., during the prescription, preparation, or administration of medications to patients. Some researchers indicate that most errors in hospitals occur at the last stage, i.e., the administration of a medicinal product to the patient [3]. As a result of the involvement of such a number of specialists in the administration of medications, starting from a doctor prescribing a medication, through nurses fulfilling the order, to a pharmacist issuing a medicinal product and approving the order, each of those specialists can trigger a situation where medication administration errors are likely to occur [4]. The literature provides many classifications of medication errors, depending on the cause and potential effect of the error, as well as modern monitoring and early prevention methods. In 2010, Westbrook, based on his observations, broke down the genesis of medication errors into procedural errors arising, for example, from the lack of patient identification prior to medication administration and clinical errors. Clinical errors include, in particular, the wrong medication dose, the omission of medications, the wrong solvent, and the wrong administration method. Westbrook confirms that 74% of medication-related incidents are errors connected with the administration of medicinal products at the procedural level, and 25% are connected with using the wrong medication dose or the wrong preparation method [5]. Direct observations of the preparation and administration of medications indicate that intravenous medication has the highest error rate. Many IV medications are identified as having a severe risk for patient harm on the high-alert medication list for acute care settings [6].

The common element connecting the environments of the OR and ICU is the range of drugs used, known as high-alert medications (HAM). Products of this type [7] should be administered with special care, all the more so because an error in dosing, preparing or administering such a medication is likely to cause the patient’s death. The American Pharmaceutical Association (APhA) included the following medicines in this group: anaesthetics, antiarrhythmic medications, anticoagulants, chemotherapeutic agents, dialysis solutions, epidural or intrathecal medications, insulin, drugs and parenteral nutrition [7]. Some HAMs have a very narrow pharmaceutical index. Warfarin, if administered too fast and in an inadequate time, may cause bleeding. Chemotherapeutic agents must be prepared, mainly based on the producer’s recommendations. Their administration method is also strictly limited to intravenous infusion. Non-observance of infusion time in such medications as insulin or potassium may even result in the patient’s death [8].

Financial and moral consequences of medication-related incidents include extended hospitalisation, increased mortality, necessary hospitalisation at the ICU, as well as increased emotional stress for the patient, the patient’s family and the medical personnel, including doctors, nurses and pharmacists [9].

Although the issue of medication errors is not new and was mentioned for the first time in the 1990s during the development of safe medicine administration rules for anaesthesiologists [10], the actual scale of the problem is still poorly known. Scientific reports confirm that owing to highly specialised treatment methods, adults hospitalised in ICUs are much more exposed to medication errors than patients in other, more general wards [11]. Patients’ health is another difficulty in analysing the seriousness of adverse events in medication administration at such wards as operating room theatres or intensive care. It is necessary to note that these are often patients with many health issues and a life-threatening condition. Logical, verbal contact with such patients is often difficult or even impossible. Therefore, it is challenging to monitor adverse symptoms upon the wrong administration of a medication [12]. The demanding and usually stressful working environment in the ward is another obstacle to analysing causes of errors in the preparation and administration of drugs. The operating theatre is an environment with a fast workflow. Anaesthesiologists and nurses focus on making quick decisions and solving problems here and now based on the patient’s clinical condition. Very frequently, during the preparation and administration of medications to the patient, the personnel have minimal opportunities to consult a hospital pharmacist, access modern IT medication administration systems or have independent double-checks [13]. In the ICU, treatment is based on wide and complex pharmacotherapy, which takes into account the dynamics of change in the patient’s clinical condition. Other factors in the ICU that favour medication errors include: an excessive workload and stress related to taking care of patients that are not able to function on their own, communication problems both with a patient’s family members, as well as between colleagues, frequent personnel shortages, intensified personnel rotation and the lack of support from superiors [14].

In theory, all issues related to the administration of medications can be prevented [3]. In the opinion of some researchers, medication error monitoring is not possible without adequate risk assessment [3,15]. Most of the research has highlighted the need to develop a system for monitoring the occurrence of ME in operating room theatres and intensive care. There are no studies that, based on original research, would present the scale of ME, as well as the conditions that favour their occurrence.

### Aim of the Study

The systematic review aimed to synthesise the published research about the number of medication errors in operating room theatres and intensive care units. A criterion was used in this review to classify an event as a medication error. As an added value, the authors of this paper developed practical recommendations that can improve both the administration of medications and patient safety, paying special attention to the methods used to evaluate the ME index rate and environmental factors that can often be modified.

## 2. Materials and Methods

### 2.1. Study Design

The literature review was conducted in the third quarter of 2023. The analysis covered original works: observational, retrospective and prospective randomised mixed-method studies. This review followed the PRISMA (Preferred Reporting Items for Systematic Reviews and Meta-Analyses) guidelines [16].

### 2.2. Search Methods

The following databases were searched: Science Direct, Ebesco, Pubmed, Ovid, Scopus and Web of Science. The following keywords were used: “medical error”, “medical errors”, “drug management”, “drug administration”, “intensive care”, “intensive cares”, “operating room”, “drug”, “drugs” or combinations thereof by use of AND or OR. There were 175 articles meeting the essential search criteria. Out of these, 13 were included in the analysis, which comprised the verification of the availability of full texts and the compliance of those texts with the inclusion criteria. Finally, the review covered articles published within the last five years from 2017 to 2023. The most recent evidence showed that no review was carried out during this period, considering that type of inclusion criteria. The review and qualification of articles for the purpose of the analysis were possible, provided that the inclusion criteria were met.

### 2.3. Inclusion and Exclusion Cryteria

The analysis covered full texts published in English. The inclusion and exclusion criteria were based on PICOS classifications referring to participants, interventions, comparisons and study design. The population included: adult population hospitalised in the ICU or operating theatre. The intervention aimed at intravenous medication errors, and the setting was an adult intensive care unit and specialist adult ICU or OR. The outcome was the rate of medical errors.

### 2.4. Literature Search and Study Selection

At the beginning of the analysis of particular databases, the authors applied a search strategy that allowed them to identify approximately 1700 scientific publications on medication errors in anaesthesiology and intensive care published in the years 1968–2023. At the review stage, 1168 scientific publications were excluded because they did not meet the basic assumptions for the publication period, which should not exceed the last 5 years. Then, the articles were verified in terms of the search criteria. The following analysis included 532 articles, 10 of which were excluded because they overlapped. 175 articles met the essential search criteria: original observational/retrospective/prospective randomised studies published in English. Finally, upon analysis regarding compliance with the PICOS criteria and the JB Institute for Research Synthesis criteria, 13 publications were subject to review.

Three investigators (K.K.-J. and W.M.-D., M.K.) screened titles and abstracts of potential studies that met inclusion criteria. After the reviewers obtained the full text of the articles, they independently extracted key information. Any doubts concerning the manuscript included in the review process were consulted. A third reviewer was also responsible for the statistical analyses and consultation of the research. Standardised protocols were used to ensure consistency in the screening and assessment process. There were no conflicts of interest between reviewers at any stage of the review process. Disagreements and doubts were resolved promptly.

The detailed list is presented in the Prisma flow chart of 2020 in Figure 1 [16]. The publications to be analysed included: a randomised study (*n* = 1), retrospective studies (*n* = 3), multicentre retrospective studies (*n* = 2), prospective observational studies (*n* = 4), a mixed-method study (*n* = 1), a cross-sectional study (*n* = 1) and database research (*n* = 1) (Table 1).

### 2.5. Data Extraction

The qualified articles meeting the inclusion criteria were analysed by three independent reviewers, being authors of the work, based on the following criteria: the author and the date, the aim of the study, type of research, the sample, materials and methods, level of evidence, the percent of incidents of ME, the recommendation for clinical practice and limitation of the study. Finally, 13 articles meeting the PICOS criteria and coming from two independent searches were included in the analysis. The detailed PRISMA checklist, based on which the search was conducted, is presented in Figure 1.

### 2.6. Ethical Aspects 

The consent of the bioethical commission was not needed to conduct the literature review due to the type of article. 

### 2.7. Assessment of the Study Quality of the Included Studies 

JBI Tools are used to critically evaluate the methodological quality of a study [28]. For this purpose, the following tool was used: JBI Critical Appraisal Checklist for Systematic Reviews and Research Syntheses, which provides a checklist with 11 criteria (Q1–Q11). The answers used are yes, no, unclear or not applicable. The results of this evaluation are presented in Table 2. 

## 3. Results

### 3.1. Characteristics of the Included Study 

The analysis revealed that individual articles are very diversified regarding observation methods. Three publications come from the United States (*n* = 3), and other countries where the studies were conducted included China (*n* = 1), Australia (*n* = 2), India (*n* = 2), Spain (*n* = 1), Turkey (*n* = 1), Great Britain with Wales (*n* = 1), Japan (*n* = 1) and the Netherlands (*n* = 1) [13,14,17,18,19,20,21,22,23,24,25,26,27]. Four of the analysed studies had a second level of research evidence, and nine of the studies had a third level of evidence. The date of identified studies (ME rate) was too heterogeneous, which is why the evidence of the study in the result section was summarised descriptively.

### 3.2. Demographic and Clinical Data 

Taking into account the area of the studies, 4 out of 13 studies analysed operating theatres and anaesthesiology. In those cases, observations and analyses of patient data concerning medication errors were conducted in operating theatres or anaesthesia intensive care units (AICUs) [13,21,24,25]. Nine publications in the review analysed medication errors at adult intensive care units [14,17,18,19,21,22,23,27,29]. One of the publications studied the frequency of medication errors both in anaesthesia and intensive care units [20]. All articles met the PICOS criteria. The analysis of particular publications indicates that the projects were quite diversified. All of the articles were original publications, and the analysis of medication errors was based on the direct observation of patients or the analysis of medical documentation and recommendations. Four articles analysed patient documentation, including the electronic history of orders [19,20,21,27]. In five studies, the analysis included ICU patients and their detailed sociodemographic data. The number of patients observed was quite diversified: from 146 to 5137 (M 1,318,857 patients) [14,17,18,20,21,22,23,26,27]. The duration of observations also varied, ranging from 7 days [14] to several months [22,26,27] or even several years [21]. Two studies on the frequency of medication errors in operating theatres were based on direct patient observation during the administration of medicinal products/medications and the analysis of reports made in the IT error reporting system simultaneously [13,24].

### 3.3. Review Findings

#### 3.3.1. Rate and Tapes of Medication Errors 

In the majority of the publications, the frequency of medication errors in OR and ICUs was estimated through the analysis of errors during the administration of medications to patients. For the operating theatre and anaesthesia intensive care unit, the medication error rate was from 7.3% to 12% [24]. In the case of ICUs, it was from 1.32% to 31.7% [19,23]. Another method applied to calculate the medication error rate involved the analysis of patient documentation by a clinical pharmacist, including, in particular, a doctor’s recommendations, vital parameter forms and descriptive data kept by nurses. In this way, it was possible both to calculate the error rate and to find out whether the error had an impact on the patient’s clinical condition [22,23,27]. The analysis conducted by Escriva et al. confirmed that medications at ICUs are usually administered intravenously. That study confirmed that 316 medication errors correspond to the global medication error index (GMEI) of 1.93%. The authors proved that there is a great risk of procedural and clinical errors both during the preparation and administration of medications. In the researchers’ opinion, a procedural error was recorded in the case of at least 74.4% of incidents and a clinical error in the case of 25%. The most frequent procedural errors included the preparation of medications without a doctor’s recommendation and the failure to identify a patient adequately before the administration of a preparation. The most frequent clinical errors included the omission of a medication dose, the administration of a preparation contrary to an infusion rate and wrong dosing [19]. Similar results were confirmed by other authors [14,20,21,22,24,25,26], who also indicated that other minor errors included the wrong marking of a medication/an infusion [13], the wrong combination of medications [22], failure to observe producer recommendations concerning the time of preparation and the administration of a medication [23], the inadequate monitoring of adverse effects [22] and the administration of a medication to a patient allergic to one of the components [25]. Results obtained by Kim et al. indicated that 38% of 4000 incidents reported as medication errors caused medium or serious damage to a patient’s health. The project also proved that 89% of errors could have been prevented [24]. 

#### 3.3.2. ME Drug Risks Area 

The studies included in the review indicated that medications at ICUs and operating rooms are most often administered intravenously [13,14,17,18,19,20,21,22,23,24,25,26,27]. Upon an in-depth analysis of the materials, the authors managed to assess which groups of medications errors are most frequent. In the case of the operating theatre and AICU, errors in the administration of intravenous medications are most frequent in the case of opioids, local anaesthetics, relaxants, insulin and anticoagulants. Other medicinal products subject to errors included relaxants and vasoactive agents, such as noradrenaline [13,24]. The analysis of the articles on medication errors proved that the group of medications subject to errors both at the preparation and administration phases in ICUs includes antibiotics, cardiovascular and nervous system drugs, narcotic analgesic drugs, anaesthetics and immune modifiers, hepatics, vasoactive drugs, sedatives and diuretics, heparin and insulin [14,20,21,22].

#### 3.3.3. Types of Environmental Factors That Lead to Medication Errors 

One of the last-but-one areas that was analysed included determining factors that lead to medication errors in ORs and intensive care units. In their study, Escriva et al. broke those factors down into four main types: “low attention of healthcare professionals to medication safety”, “lack of professional communication and collaboration”, “environmental determinants” and “management determinants” [19]. In the case of the operating theatre, rapid workflow constituted the most frequent environmental factor increasing the risk of medication errors related to work culture. Other factors specified by other authors included: pressure of time, fatigue, multiple people and lack of proper communication [24] and lack of opportunities for the development and improvement of knowledge on the administration of medications at the ward [25]. The analysis made by Kim et al. [24] also indicated that medicinal products were marked and stored wrongly, and look-alike medications could contribute to errors. In ICUs, apart from individual factors related both to the personnel and patients, it was confirmed that medication errors result from the lack of relevant communication between teams, inadequate work organisation, including interruptions in medication administration, the misunderstanding of the definition of an error and error consequences, as well as the lack of employee loyalty to the organisation [27]. Studies made by Tully et al. [14] proved that the risk of medication errors is greater due to the organisation of hospital work, including the number of ICU beds exceeding 25. The referral level of hospitals is also an important factor, as highly specialised units that perform complicated medical procedures are more exposed to errors [14].

#### 3.3.4. Action Taken to Prevent Medication Errors 

Each of the 13 publications analysed hereunder presents important recommendations aimed at decreasing medication error rates at hospitals, in operating rooms and in intensive care units. The researchers point out a need to: both introduce IT systems for everyday clinical purposes, as well as safe infusion pumps [24], introduce medical procedures for the preparation of medications [20], increase the presence of a clinical pharmacist [13], mark syringes adequately and double-check medications prepared and administered [24,25]. Two articles that analyse ICU activities state that intravenous medications must be subject to double-checking. Three of the publications confirm that it is necessary to increase the participation of a clinical pharmacist in administering medications and verifying recommendations [17,22,23]. One of the studies turns attention to implementing safe medication preparation and administration procedures and a checklist of antibiotics administered to patients [19]. Four studies confirm that introducing new technologies at ICUs, including an electronic system of doctors’ recommendations, advanced error reporting systems and pumps with a medication library, can reduce medication errors significantly [18,19,20,21]. Other authors confirm that everyday rounds [14] and adequate management of patient check-in and check-out times [14] significantly generate medication errors. 

## 4. Discussion

Owing to progress in medicine, in terms of medicinal products, and the introduction of new technological solutions and intuitive and smart medical equipment to support medical personnel, incidents in the administration of medicines can be better and better monitored [15]. Efforts are based on the development of IT programmes reporting medication errors and on a practical approach to improving patient safety during surgery and hospitalisation at the ICU [29,30]. The aetiology of errors in the area of the administration of medications is highly intricate, and the studies analysed and reviewed hereunder confirmed that this is a multidimensional problem [15]. Although operating room theatres and intensive care units combine high specialisation, innovative solutions and advanced equipment, they are not an exception. The frequency of medication errors is relatively high owing to the broad scope of medications used there; however, the error rate does not differ from general rates [31]. Based on the articles reviewed hereunder, it was possible to identify the number of medication errors, the medication error rate in operating room theatres and in intensive care units, the nature of errors and factors that have a significant impact on the occurrence of such errors.

### 4.1. Types and Rate of Medication Errors 

During the analyses, it was confirmed that a medication error can be procedural, i.e., resulting from failure to identify a patient properly prior to the administration of a medication or the wrong preparation or storage of a medicinal product, and clinical [13,14,17,18,19,20,21,22,23,24,25,26,27]. In the latter case, the omission of a medication or the administration of the wrong preparation was the most frequent clinical issue. In addition, based on the analysed materials and reports made by other scientists, it can be confirmed that none of the above types of errors is isolated [3]. Usually, which is confirmed by most researchers, both types of errors exist at the same time and contribute to a further cascade of unexpected events, including adverse drug reactions (ADR) [31]. The circulation of medicinal products in the hospital environment also contributes to medication errors. It is necessary to note, however, that opinions in this area differ. Some researchers state that most medication errors occur during the preparation and administration of medicinal products to patients [20,27,31]. At that stage, the errors are mostly caused by nurses [19]. Other analyses confirm that errors may occur at each administration stage, starting from the delivery of the wrong product from the hospital pharmacy to the issue of the wrong recommendation [27,28,31]. In the analysis of the medication administration error index, calculations made by the researchers indicated that the index was for the ICU and for the operating theatre. These are high values; however, they do not differ from general WHO error indexes [1]. 

### 4.2. Drug Risks Area 

The analyses confirmed that the most frequent groups of medications subject to errors include HAMs, such as vasoactive drugs and other drugs influencing the blood circulation system, pain killers, antibiotics, heparin, insulin, CNS medication and proton pump inhibitors. Only 4 out of 13 studies analysed hereunder did not include any information concerning medications [13,17,18,23]. The analyses are consistent with observations made by other authors. Therefore, to avoid errors in the analysed area, it is recommended to mark medications and develop protocols for preparing such medicines in the hospital environment [32]. 

### 4.3. Types of Factors That Lead to Medication Errors 

The analysed material indicated that factors that can lead to medication errors include the hospital’s size and the number of beds [14]. Other factors included significant workload, tiredness and overload of nurses and doctors, the incorrect marking of syringes and medications and inadequate communication within the team [19,21]. It was also reported that other distractors leading to medication errors include the patient’s critical condition [33], as well as an interruption to the preparation of medication by colleagues or noises in the working environment (monitor alarms, telephones) [34,35,36,37]. Estok [33] observed that the haemodynamic instability of a patient, or bleeding during the administration of infusion medications, can increase stress in medical personnel and generate a greater risk of errors. Observations made by Kim et al. proved that medication errors are much less frequent among patients classified in ASA 4–5, 80+ patients or patients between 1 and 16 years of age, with a BMI over 40 [24]. Other significant problems mentioned by numerous publications that were not included in the review were the medical personnel’s poor awareness of factors leading to medication errors and the lack of knowledge on the adequate preparation and administration of medications, including, in particular, HAM medications [8,38]. Night work, double shifts or disturbances to sleep and the day rhythm of the night personnel are other important factors mentioned by researchers [35,39].

### 4.4. Safety Recommendations for Specialists and Hospital Managers 

Safety in the management and administration of medications can be provided through the optimisation of processes in the working environment, including the development of prevention strategies, the elimination of risk factors and distractors, as well as the monitoring of MEs. The articles analysed hereunder proved that it is reasonable and effective to make changes in the hospital environment, starting from simple solutions, such as the proper marking of syringes and the double-checking of medications at the preparation stage, through to modern technologies, reporting systems and smart infusion pumps [13,14,17,18,19,20,21,22,23,24,25,26,27]. The results of the analyses are consistent with other authors’ publications. Apart from the aforementioned solutions, Nunes et al. suggest that the relevant marking of medications based on a colour coding system should be implemented. In addition, they turn attention to the necessary use of homogeneous equipment at anaesthesia tables, medications and the equipment used to prepare medications at each station. In the article, the experts also mention that it is necessary to introduce module systems for drug preparation area, increase the participation of a clinical pharmacist in the preparation of medications, and implement a colour-coding system on trays [40]. Observations made by Laxton et al. confirmed that the anaesthesia environment is ready to use rainbow trays in everyday practice [41]. Although there is a need to develop a tool, Nanji K.C and her research team recommend the use of a perioperative clinical decision support application that can reduce the incidence of perioperative MEs, including dosing errors, errors of omission, monitoring errors and wrong medication errors [42]. In the case of HAM medications, the medical personnel should receive ready-to-use preparations from the hospital pharmacy (in particular, heparin, insulin, antibiotics or vasoactive drugs). Similar solutions should also be applied to epinephrine. Highly concentrated medications, e.g., potassium, which, if prepared incorrectly, could cause the patient’s death, should be available in a ready-to-use form [43] or as an infusion [44]. In the marking systems, it is important to properly mark both medications and all ports to which medications can be administered with colours, including, in particular, an epidural entrance (marked in yellow) or an arterial entrance (marked in red). The proper storage of products is also essential. Hypertonic drugs, heparin, concentrated glucose or epidural solutions should be stored separately [21,43,45,46]. In ICUs, other solutions frequently mentioned as improving safety include the implementation of barcodes, operating infusion pumps from the computer portal and increasing the participation of a clinical pharmacist during rounds [43]. Regarding environmental factors, it seems necessary not only to monitor factors leading to medication errors but to also prepare schedules adequately without cumulative hours, hold training sessions on pharmacotherapy and support personnel, particularly at the beginning of their professional careers [3,15]. Other researchers also point out a need to increase the participation of nurses in the planning of patient check-in and check-out because, as confirmed by observations made by other researchers, these are critical moments, and nurses’ needs and opinions are not always taken into consideration in this process [47]. 

## 5. Conclusions

On the basis of the articles included in the review, the conclusion that the number of MEs in ORs and in ICUs is high can be drawn. The values presented herein do not differ, however, from the general ME index for medical centres, as calculated by the WHO.

## 6. Implication for Practice

The analyses indicated that new technologies and innovative solutions are very important; however, it is also important to make simple and reasonable changes, such as increasing the amount of training on pharmacotherapy and new medications used in hospitals, improving the awareness of the nature of medical errors among medical personnel (nurses, doctors, clinical pharmacists), as well as reducing workload and avoiding the accumulation of shifts during the working week. 

## 7. Limitation of the Study

This review has a few limitations. Firstly, although an inclusive approach was used and many databases were checked, some publications may have been missed. Secondly, there were no ideal tools to calculate the index of medication administration error and estimate the factors that lead to the incidents of mistakes in hospital environments. Some original research included in the study analyses the incidence of medication error using retrospective analysis, and another research focuses on that problem during real-time observation and prospective tools.

## Figures and Tables

**Figure 1 medicina-61-00369-f001:**
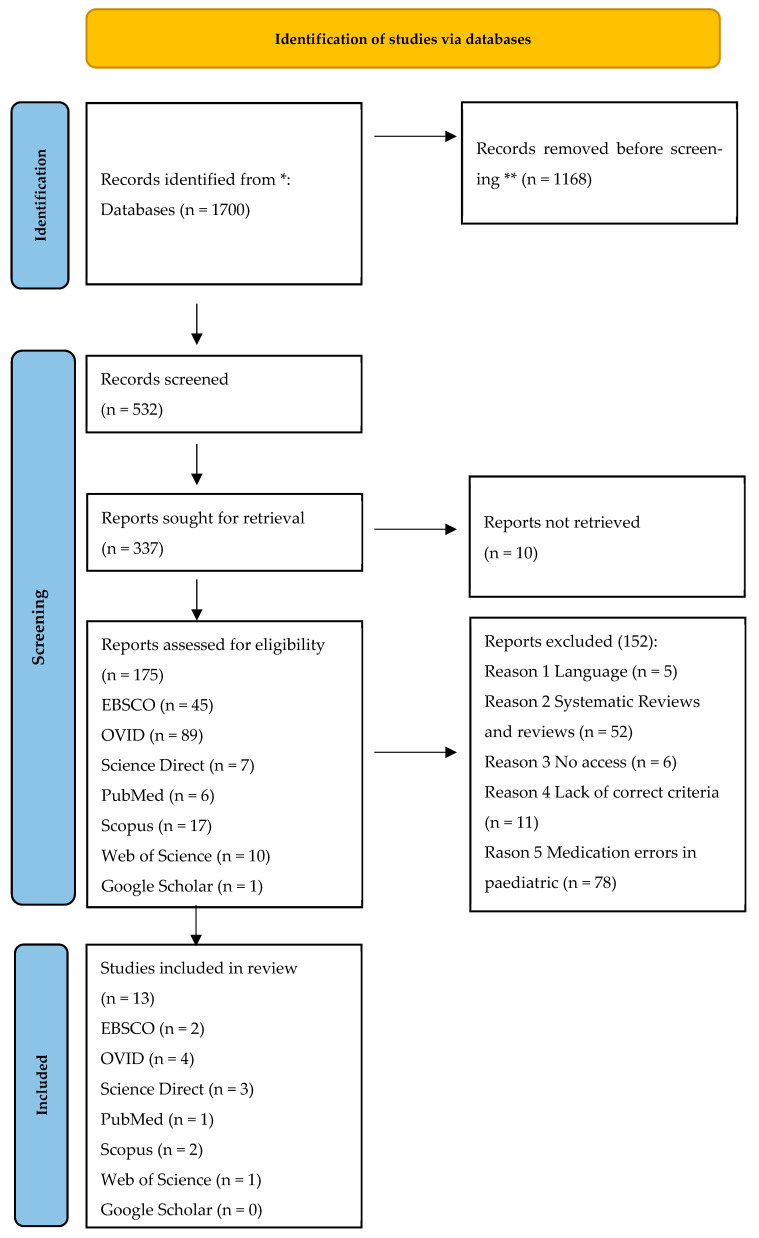
Prisma flow chart. Legend: *—Records identified from 1700 database; **—records removed because they did not meet inclusion criteria according to the year of publication.

**Table 1 medicina-61-00369-t001:** Synthesis of qualitative findings for a literature review in the field of medication error in the operating room theatre and intensive care units.

Athor and the Date	The Aim of the Study	Type of Research	Sample	Materials and Methods	The Percent of Incidente of ME	Area	The Level of Evidence	Recomendation for Practice	Limitation of the Study
Chalasani and Ramesh, 2017 [17]	To determine the incidence, causes, patterns and outcomes of medication errors (MEs) in the intensive care unit.	Original research	5137 patients	The study was conducted using a newly established ME reporting centre in the ICU of a tertiary care university hospital. The data were collected for 6 months.	The incidence of ME was 5.6%.	ICU	II	To reduce MEs, reporting systems should be designed and education programmes implemented, assuring a non-punitive and reward-based system.	The authors did not declare.
Liao et al., 2017 [18]	Compare the number of MEs before and after EHR implementation in the MICU, with additional evaluation of error severity.	The prospective observational study	673 ICU patients	The observational study was lead during four periods: August–September 2010 (preimplementation; period I), January–February 2011 (2 months postimple-mentation; period II), August–September 2012 (21 months postimplementation). Patients that stayed on the unit less than 24 h were excluded from the study.	There was a statistically significant increase in the number of MEs per 1000 patient days during time periods II (n = 2592; *p* < 0.001) and III (n = 2388; *p* = 0.0023) compared to baseline (n = 1972). However, over time there was a significant reduction in medication errors during period IV compared to baseline (n = 1669; *p* = 0.0008).	ICU	II	The implementation of an electronic health record system in 2 observations decreased the amount of medication error.	One limitation of the study is instrumentation during the chart review process and after the implementation. Another limitation was that researchers did not include a review board to adjudicate each error for severity and categorisation.
Escrivá Gracia et al., 2019 [19]	This study identifies the main medication errors that occur in the ICU at a general hospital in the city of Valencia (Spain).	Mixed multimethod study	2634 medications administered.	The mixed-method study was led. The study structured into the three phases that involve collecting quantitative and qualitative data. The analysis of qualitative data was conducted in the second phase, quantitative data in the third phase.	The error rate in the transcription and transition process was 1.32%.	ICU	III	One way of preventing ME is analysing the key variables that lead to the ME and developing a prevention strategy.	Small analysis sample, error analysis methods used in the study.
Härkänen et al., 2019 [20]	Analyse medication administration errors reported in acute care resulting in death.	Retrospective study	The total number of incidents extracted was 517,384. Of that, 229 of them (0.4%) resulted in patient death.	NRLS extracted the data (medication administration errors reported to the NRLS between January 2007 and December 2016) in December 2017.	Error incidence rate in the ICU was 7.9% (n = 18), 4.4% in the operating room (n = 4), 0.9% in the recovery room (n = 2)	ICU, operating room theatre, other specialized units	III	To prevent the most serious medication errors, interventions should focus on: avoiding dose omissions and the administration of drugs for patients over 75 years old.	The self-reporting system is the possible weakness of the research.
Tully et al., 2019 [14]	To determine the point prevalence of medication errors at the time of transition of care from an ICU to a non-ICU location. To assess error types and risk factors for medication errors during the transition of care.	Multicentre retrospective study.	985 patients transferred from an ICU to a non-ICU location	The participating pharmacists collected and recorded deidentified data in a secure, web-based application (REDCap 7.6.9, Vanderbilt University, Nashville, TN) that was developed and maintained by The Johns Hopkins University School of Public Health. Pharmacists were recruited via email communication to participate in data collection.	Among patients with a medication error, an average of 1.88 errors per patient occurred.	ICU	III	Factors associated with decreased odds of error included daily patient care rounds in the ICU and orders discontinued and rewritten at the time of transfer from the ICU.	Authors could not control potential interrater variability in reporting a medication error system. Secondly, researchers could not exclude the possibility of variation in self-reporting of errors.
Bosma et al., 2021 [21]	To characterize prescribing, monitoring and medication transfer errors that were voluntarily reported in the ICU, in order to reveal medication safety issues.	Retrospective data analysis	Medical record	All reported ME in 2016 and 2017 occurring in the ICU were extracted from the hospital IRS databases.	Type of error: errors related to the choice of medicine (25%), errors related to dosing, frequency and duration of therapy (15%) and errors related to the medication surveillance (1%).	ICU	III	Prescribing errors can be prevented by medication safety practices like electronic prescribing applications and the daily attendance of a pharmacist during patient rounds.	One of the limitations of the study is the voluntary reported ME might be under-reported.
Aghili and Neelathahalli Kasturirangan, 2021 [22]	To evaluate the incidence rate, type and severity of medication errors and preventable adverse drug events (ADEs). To assess the impact of the implementation of interventions recommended by the clinical pharmacist.	Prospective study	There were 146 patients out of 228 patients/ 1000 days of hospital chart observations. The study was conducted from November 2017 to January 2019 in the medical ICU.	The clinical pharmacist performed a combination of medication error detection methods, which included medication chart review, patient monitoring until discharge/death and attending medical rounds. Detected medication errors were intervened with prescribers. Patients were divided into two groups, A and B.	A total of 271 medication errors with an incidence rate of 122.62 per 1000 patient hospital- days were detected.	ICU	II	The clinical pharmacists should participate in the process of pharmacotherapy. It would be worthwhile to indicate the need for dissemination of drug information among pre- scribers.	Some medication errors and preventable ADEs may have been missed. Researchers identified medication errors at the prescribing stage of the medication use process.
Ayhan et al., 2022 [23]	To reduce DRPs and associated costs with clinical pharmacist’s (CP) recommendations.	Prospective nonrandomised study	There were 146 patients included in the study and a total of 1061 drug related problems were detected.	The medication of 146 patients was included in the analysis and evaluated by clinical pharmacists.	The most common problem of drug-related problems in those 3 periods of observations were: inappropriate combination of drugs (31.7%), high dose of drugs (12.4%) and errors in dose timing instructions (9.2%).	ICU	II	Clinical pharmacist should cooperate with other members of the health team (nurses and physicians). The routine participation of clinical pharmacists in patient rounds is helpful to recognize mistakes, prepare more suitable treatment plans and reduce the cost of hospitalization.	The acceptance rate of the recommendations for drug-related problems (DRPs) was high, and the rates of complete implementation of the recommendations were not sufficient.
Kim et al., 2022 [24]	To analyse the incidents related to medication error in the first 4000 cases reported to webAIRS and thereby to advance further webAIRs’ overall objective of improving patient safety by learning from incident reporting.	Original research	There were 4000 reports analysed and 71 reports were excluded from the analysis. In the remaining 3929, 687 reports had ‘medication’ as their main category.	There were 4000 reports from the webAris system analysed.	Of the first 4000 reports to webAiris, 12% involved medication error, 89% of these were considered preventable and 38% caused moderate or severe harm to a patient.	Operating room and post anaesthesia care unit (PACU).	III	One of the starting points to reduce ME is: investments in technology and staff training, clearly labelled syringes, double-checking of drugs, especially those with low-volume, high-risk routes of administration (notably spinal and epidural), training in teamwork and communication among medical staff and developing the proper work culture to reduce staff fatigue.	One of the limitations of the study was the type of reporting system—voluntary. Some medication errors might be missed.
Stipp et al., 2022 [13]	To evaluate perioperative medication-related incidents (medication errors (MEs) and/or adverse medication events (AMEs).	Observational studies	There were 237 anaesthesia clinicians and 277 observations involving 3671 medications.	There were 277 observations involving 3671 medication administrations and 193 MEs and/or AMEs were observed.	Error incidence rate of 5.3%.	Operating room theatre and anaesthesia	III	There are 4 strategies that may reduce the incidence of ME: pharmacy reconstituted medications, barcode-assisted medication administration, standardised medication organisation and incident reporting.	The first limitation was the period of the study from November 2013 to June 2014. The study does not account for safety provisions that have been implemented since the specified timeframe.
Suzuki et al., 2022 [25]	To prevent drug-related medication errors in the operating room by clarifying the association between the medication error category with related drugs and contributing factors.	N/A	The total number of analysed cases from 727 was 541.	Researchers used data from the Japan Council for Quality Health Care’s open database on the web. They researched the medication error category, related drugs and contributing factors. They classified each medication error category.	N/A	Operating room theatre	III	Standardisation procedures, dissolved IV medication, proper labelling of the drug or having the drug ready to use in syringes will reduce the incidence of omitted medications. The use of checklists will improve the correct administration of antibiotics.	Reporting bias cannot be excluded because the reports of medication error cases from the database used in this study are spontaneously and voluntarily reported.
Wang et al., 2022 [26]	To describe the incidence and types of medication errors occurring during the transfer of patients from the intensive care unit (ICU) to the non-ICU setting. To explore the key factors affecting medication safety in transfer care.	Multicentre, retrospective, epidemiological study.	A total of 1546 patients were included from three tertiary hospitals in Anhui Province.	Procedural and clinical medication error. The three most common types of medication errors were route of administration (37.85%), dosage (17.99%) and frequency (9.23%).	The general incidence of ME.	ICU	III	More than half of ICU patients experienced medication errors during the transition of care. It is important to reduce the risk of drug use in the care transfer process.	One of the limitations of that study was that the medication orders were collected only within 24 h before and after ICU transfer. Medication errors occurring outside of the prescribed time period could not be detected.
Ottosen and Bucknall, 2023 [27]	1. Determine the frequency and severity of medication errors reported in the incident management reporting system. 2. Examine the antecedent events, their nature, the circumstances, risk factors and contributing factors leading to medication errors. 3. Identify strategies to improve medication safety in the ICU.	Original paper	A total of 162 medication errors were reported.	Data were collected from the incident report management system and electronic medical records over a 13-month period from a major metropolitan teaching hospital ICU.	There were 150 eligible for inclusion.	ICU	III	Improving administration-checking procedures would prevent the occurrence of many medication errors. A two –person checking procedure should be implemented to reduce the incidents of medicine errors.	One of the limitations of the study is the use of a single method at a single time.

Legend: webAIRS—system to analyse medication error; ADR—adverse drug reaction; IRS—incident reporting system; MTE—medication transfer error; EHR—electronic health records; MICU—medical ICU; ICU—intensive critical unit; National Reporting and Learning System—NRLS; MAE—medication administration error; CP—clinical pharmacists; DRP—drug-related Problem.

**Table 2 medicina-61-00369-t002:** Critical appraisal results for included studies.

Author, Year	Q1	Q2	Q3	Q4	Q5	Q6	Q7	Q8	Q9	Q10	Q11
Chalasani and Ramesh, 2017 [17]	Y	Y	Y	Y	Y	Y	Y	n/a	n/a	Y	Y
Liao et al., 2017 [18]	Y	Y	Y	Y	Y	Y	Y	n/a	n/a	Y	Y
Escrivá Gracia et al., 2019 [19]	Y	Y	Y	Y	Y	Y	Y	U	U	Y	U
Härkänen et al., 2019 [20]	Y	Y	Y	Y	Y	Y	Y	n/a	n/a	Y	Y
Tully et al., 2019 [14]	Y	Y	Y	Y	Y	Y	Y	n/a	n/a	Y	Y
Bosma et al., 2021 [21]	Y	Y	Y	Y	Y	Y	Y	n/a	n/a	Y	Y
Aghili and Neelathahalli Kasturirangan, 2021 [22]	Y	Y	Y	Y	Y	Y	Y	n/a	n/a	Y	Y
Ayhan et al., 2022 [23]	Y	Y	Y	Y	Y	Y	Y	n/a	n/a	Y	Y
Kim et al., 2022 [24]	Y	Y	Y	Y	Y	Y	Y	n/a	n/a	Y	Y
Stipp et al., 2022 [13]	Y	Y	Y	Y	Y	Y	Y	n/a	n/a	Y	Y
Suzuki et al., 2022 [25]	Y	Y	Y	Y	Y	Y	Y	n/a	n/a	Y	Y
Wang et al., 2022 [26]	Y	Y	Y	Y	Y	Y	Y	n/a	n/a	Y	Y
Ottosen and Bucknall, 2023 [27]	Y	Y	Y	Y	Y	Y	Y	n/a	n/a	Y	Y

Y—Yes, N—No, U—Unclear, n/a—not applicable. Q1: Was the review question clearly and explicitly stated? Q2: Were the inclusion criteria appropriate for the review question? Q3: Was the search strategy appropriate? Q4: Were the sources and resources used to search for studies adequate? Q5: Were the criteria for appraising studies appropriate? Q6: Was the critical appraisal independently conducted by two or more reviewers? Q7: Were there methods to minimise errors in data extraction? Q8: Were the methods used to combine studies appropriate? Q9: Was the likelihood of publication bias assessed? Q10: Were recommendations for policy and/or practice supported by the reported data? Q11: Were the specific directives for new research appropriate?

## Data Availability

No new data were created or analysed in this study.

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
