# Peer review of "Understanding Medication Errors in Intensive Care Settings and Operating Rooms—A Systematic Review"

_medicina, 2025, doi:10.3390/medicina61030369_

Round 1
Reviewer 1 Report
Comments and Suggestions for Authors
I am unsure of the novelty or additional information this paper adds. They do not reference British Journal of Anaesthesia, 127 (3): 458e469 (2021), which was also a literature review that extracted more papers. Moreover, that paper categorised error by type.
There is some novelty in that these authors extend their analysis to ICU. However, they have missed very many key articles such as works by Nanji et al. so the basis of their study may not have been on a robust search or sieving criteria. Their Methods do not specify aspects like exclusion criteria, years of search, or indeed what their specific goals were by this review.
Comments on the Quality of English LanguageThere are places where it can be improved.
Author Response
Dear Sir or Madam:
Thank you very much for your helpful review. We have studied the comments and suggestions carefully and revised our paper accordingly. Below are our point-by-point responses to the general and specific comments. We hope the revisions are acceptable and that our responses adequately address your observations. I appreciate your consideration. All corrections have been highlighted in red. The manuscript was once again revised to improve its English translation.
Reviewer 1 |
|
Suggestions/comments |
Response |
I am unsure of the novelty or additional information this paper adds. They do not reference the British Journal of Anaesthesia, 127 (3): 458e469 (2021), which was also a literature review that extracted more papers. Moreover, that paper categorised error by type.
There is some novelty in that these authors extend their analysis to ICU. However, they have missed very many key articles such as works by Nanji et al. so the basis of their study may not have been on a robust search or sieving criteria.
Their Methods do not specify aspects like exclusion criteria, years of search, or indeed what their specific goals were by this review.
|
Thank you for your volunable comment. Unfortunately, we con not agree wiet the statement that our manuscript does not add anything new in the topic. The manuscript that was suggested is, without any doubt, very interesting because it gives a review of the evolution of methods used to estimate the rate of medication errors in hospital settings (Webster C.S. The evolution of methods to estimate the rate of medication error in Anaesthesia.British Journal of Anaesthesia, 127 (3): 346e349 (2021) doi: 10.1016/j.bja.2021.06.008 ). Our review is taking into consideration the estimation of the number of medication errors in the anesthesiology and intensive care environment. Considering the inclusion of criteria, we include studies from the last 5 years and evaluate their evidence strength. The recommended article gives the feedback that there is no sense in comparing medication errors in adult anaesthesia vs. pediatric patients in the wide heterogeneity of studies. The results will be false. That is why in our study on inclusion criteria, we Focus only on the adult units, excluding manuscripts describing medication errors in the pediatric unit.
The work of Nanji KC, Garabedian PM, Shaikh SD, Langlieb ME, Boxwala A, Gordon WJ, Bates DW. Development of a Perioperative Medication-Related Clinical Decision Support Tool to Prevent Medication Errors: An Analysis of User Feedback. (Appl Clin Inform. 2021 Oct;12(5):984-995. doi: 10.1055/s-0041-1736339.) is very interesting but in detailed it is done in the simulation. Patients included into the study are not real patients but members of medical Staff (nurses, doctors) that is why we could not include that manuscript in the review process. Nevertheless, the article recommended by the Reviewer is undoubtedly very interesting, which is why we enclosed it in the Discusion section. “…Observations made by Laxton et al. confirmed that the anaesthesia environment is ready to use rainbow trays in everyday practice [42]. Although there is a need to develop a tool, Nanji K.C and her research team recommend the use of a perioperative clinical decision support application that can reduce the incidence of perioperative MEs,1 including dosing errors, errors of omission, monitoring errors, and wrong medication errors [43].”
Dear Reviewer, thank you for your suggestion, but we can not agree with your opinion. The methods (according to the third reviewer's opinion) are described in detail in the text (from line 111 to line 153). Below, you will find the list with detailed descriptions: · Inclusion and exclusion criteria in subsection 2.3 (lines 130-136) · Year of search (lines 124-125) – „...Finally, the review covered articles published within the last five years from 2017 to 2023. The most recent evidence showed that no review was carried out during this period, considering that type of inclusion criteria. · The goals of review (lines 127-133)
|

Reviewer 2 Report
Comments and Suggestions for Authors
- Typo
1. Title
o "Type of the Paper (Article" → "Type of the Paper (Article)")
2. Abstract
o "purpse" → "purpose"
o "mediacation" → "medication"
o "operatng" → "operating"
3. PRISMA Flow Chart Caption
o "cryteria" → "criteria"
o "acording" → "according"
o "Rason" → "Reason"
4. Data Extraction
o "observatonal" → "observational"
o "prospective randomised mix-method" → "prospective randomized mixed-method"
5. Results
o "cryteria" → "criteria"
o "comparisons" → "comparisons"
6. Discussion
o "commmunication" → "communication"
o "medicatioon" → "medication"
o "enviromental" → "environmental"
7. Other, please check:
o "increse" → "increase"
o "syryngies" → "syringes"
o "recomendation" → "recommendation"
o "comminucation" → "communication"
o "incydents" → "incidents" - The manuscript includes a definition of medication error, but it does not explicitly provide an
operational definition used in this systematic review. To improve clarity of definition of medication
error used in this review, you can clearly stated :
- the criteria used in this review to classify an event as a medication error
- if there’s only certain categories of medication errors are analyzed (e.g., administration
errors only, or also including prescribing and dispensing errors)
and add how variations in definitions across different studies are addressed in this review. - PRISMA Flow Chart
- In Eligibility Check stage. The number given (162) does not match the expected number
(175).
- Possible reasons for this discrepancy: An error in calculating the number of reports assessed for eligibility.
o Inconsistent data between the methodology section and the PRISMA Flow Chart.
- Please Recheck the number of articles and adjust the numbers in the PRISMA Flow Chart to match the data reported in the text. - Appraisal for good systematic review:
- Clearly defined research question (PICO framework) -ok
- Clear inclusion and exclusion criteria -> but some details (such as justification for excluding pediatric studies) could be clearer.
- Comprehensive search strategy - suggesting add search terms and Boolean operators in more detail for reproducibility.
- Systematic literature search - please check the number
- Study screening and selection process - there’s no clearly stated if there’s conflicts between reviewers.
- Critical appraisal of included studies - Though already mentioned JBI tools-use for quality assessment, it would be better if the results of the appraisal could be discussed more thoroughly in the text.
- Data extraction and synthesis - ok
- Transparent reporting (PRISMA guidelines) - wondering a summary table of key study characteristics would improve clarity.
- Discussion and implications - see next - Result and Discussion:
a. There is some repetition between these sections. The distinction between reporting results
and interpreting them is not always clear. b. Suggestions:
o Avoid Redundancy
o The Results section should strictly present findings without interpretation (e.g.,
statistical data, trends, extracted study characteristics).
o The Discussion should then analyze these findings, explain their significance,
compare them with previous research, address potential biases and enhance the Interpretation o Arrange the Results in a structured way, and improve logical flow.
o Each key result should only be stated once in the Results and then elaborated on in the Discussion rather than repeated.
Author Response
Dear Sir or Madam:
Thank you very much for your helpful review. We have studied the comments and suggestions carefully and revised our paper accordingly. Below are our point-by-point responses to the general and specific comments. We hope the revisions are acceptable and our responses adequately address your observations. I appreciate your consideration. All corrections have been highlighted in red. The manuscript was once again revised to improve its English translation
Reviewer 2 |
|
Suggestions/comments |
Response |
|
Thank you for your suggestion and comment. We agree with your suggestion. All spelling mistakes were corrected. We strongly regret that we did not avoid such errors.
Thank you for your valuable comments. We revised the manuscript once again and added the definition of medication error to improve its clarity. – “…The systematic review aimed to synthesise the published research about the number of medication errors in operating room theatres and intensive care units. The criteria used in this review to classify an event as a medication error.”
Dear Reviewer, Thank you for your suggestion. We have checked the Prisma flow Chart and the number of articles included in the diagram and text. We strongly regret that we did not avoid such mistakes.
Thank you for your suggestions and comments. We agree and follow your recommendation. The PICO framework was clearly defined The text contained more detailed information about exclusion and inclusion criteria. We rechecked the number of articles included in the reviewer process. We expand subtitle 2.4 2.4. Literature Search and study selection. “….Three investigators (KK-J and WM-D, MK) screened titles and abstracts of potential studies that met inclusion criteria. After the Reviewers obtained the full text of the articles, they independently extracted key information. Any doubts concerning the manuscript included in the review process were consulted. A third Reviewer was also responsible for the statistical analyses and consultation of the research. Standardised protocols were used to ensure consistency in the screening and assessment process. There were no conflicts of interest between reviewers at any stage of the review process. Disagreements and doubts were resolved promptly…” Unfortunately, we can not agree to describe the JBI tools' results in the text because that opinion contradicts another reviewer’s suggestion.
Dear Reviewer, Thank you for your guidelines and support. We revised the discussion section once again, explaining the significance of the findings and comparing them to other results from that area. We also added the article to the discussion text, as suggested by the first Reviewer. We try to avoid redundancy and try in the result section to focus mostly on the description of findings. It was not easy to focus only on the data characteristic because, as we stated in the subsection 3.1 3.1. Characteristics of the Included Study The articles included in the review process were heterogeneous. The authors of the included articles used different tools to calculate the medication error-index, which is why we had to explain how they reached that kind of result.
We agree with your suggestion. The discussion was once again up-to-date, and some of the key points results were excluded from that part of the manuscript.
|

Reviewer 3 Report
Comments and Suggestions for Authors
Overall the main manuscript is of interest in this fied of research. the methods are accurate but should be reported a forest plot for each article proposed, add the quality of the paper in according to the robust approach. the statistical approach should be deeply revised by an expert in particular regarding the sensibility and specificity, accuracy of the study. This is a major issue that should be addressed that limited the impact of the paper.
In the discussion the authors should take into account how is possible to overcome this issue (Simualtion, bundle?). this point can offer an interesting perspective for the readers
I reccomend a revision by expert statiscian about this paper
Comments on the Quality of English Language
the manuscript should be revised by english mother tongue with experience in scientific literature
Author Response
Dear Sir or Madam:
Thank you very much for your helpful review. We have studied the comments and suggestions carefully and revised our paper accordingly. Below are our point-by-point responses to the general and specific comments. We hope the revisions are acceptable and our responses adequately address your observations. I appreciate your consideration. All corrections have been highlighted in red. The manuscript was once again revised to improve its English translation
Reviewer 3 |
|
Suggestions/comments |
Response |
Overall the main manuscript is of interest in this fied of research. the methods are accurate but should be reported a forest plot for each article proposed, add the quality of the paper in according to the robust approach. the statistical approach should be deeply revised by an expert in particular regarding the sensibility and specificity, accuracy of the study. This is a major issue that should be addressed that limited the impact of the paper.
In the discussion the authors should take into account how is possible to overcome this issue (Simualtion, bundle?). this point can offer an interesting perspective for the readers I reccomend a revision by expert statiscian about this paper
|
Thank you for your suggestion and comment. We have, accordingly, done statistician revision after we included all 13 manuscripts in the review process. Unfortunately, because of the vast heterogeneity of studies (different types of data), meta-analyses were impossible to do. This was the most common reason data was described in the descriptive characteristic. That information was pointed out in the result section (lines 239-241)- „…. The date of identified studies (ME rate) was too heterogeneous, which is why the evidence of the study in the result section was summarised descriptively.”
Thank your comment. The discussion was revised once again. We take into consideration your suggestion about the simulation. According to Reviewer 1 opinion, we add an article about new technologies in simulation (lines 419-423)- „…Although there is a need to develop a tool, Nanji K.C and her research team recommend the use of a perioperative clinical decision support application that can reduce the incidence of perioperative MEs,1 including dosing errors, errors of omission, monitoring errors, and wrong medication errors.”
|

Reviewer 4 Report
Comments and Suggestions for Authors
Dear authors, I have attached the comments to your work. Congratulations!
Dear Authors,
The manuscript ‘Understanding Medication Errors in Intensive Care Settings and Operating Rooms - A Systematic Review with Recommendations for Evidence-Based Practice’ is undoubtedly a contribution to updating knowledge in this specific area. In my opinion it can contribute to improve practical interventions in health services.
The study demonstrates consistency throughout its sections, including the introduction, objectives, materials and methods, results, discussion, conclusions and implications for practice.
It's just a suggestion, but because I think the title is too long, why not just : Medication Errors in Intensive Care Settings and Operating Rooms - A Systematic Review.
Systematic literature reviews should provide implications for practice. They are essential for synthesizing existing evidence, identifying gaps in knowledge, and offering data-driven guidance for informed decision-making.
It is suggested that acronyms should not be used for the first time without being preceded by the word written out in full with the acronym in brackets (ME Index and WHO), which can be seen in the abstract.
The method has many positive aspects, including a careful selection of articles, clear definitions of inclusion criteria, and the use of tools for assessing study quality.
The text written in lines 116 and 117 appears in red, please correct it.
Applying clear criteria, such as a five-year limit for the publications included, is a good practice that helps to focus the review on the most current and relevant literature. It is suggested that the justification for this time limit, could be this (the most recent evidence, considering that no review was found carried out in this period).
In the prisma flow chart, when we add up the articles from the different databases, it makes a total of 15 articles and not 13, which is the number mentioned by you for the full analysis. Please check
The use of Joanna Briggs Institute tools to critically evaluate the methodological quality of included studies strengthens the review. Congratulations.
Results are adequately described.
Between lines 334 and 337 of the article, you state: “Five of the studies confirm that the introduction of new technologies at ICU’s including an electronic system of doctor’s recommendations, advance error reporting systems and pumps with a library of medications, can reduce the number of medication errors significantly”, but only 4 of the 5 studies are cited. Please confirm that.
It presents a discussion where the results are confronted with other studies, as well as the healthcare impact.
It also presents the implications and limitations of the study, as well as recommendations for practice.
Conclusion is consistent with the objective.
The references are adequate and in sufficient number.
Congratulations on your work.
Author Response
Dear Sir or Madam:
Thank you very much for your helpful review. We have studied the comments and suggestions carefully and revised our paper accordingly. Below are our point-by-point responses to the general and specific comments. We hope the revisions are acceptable and our responses adequately address your observations. I appreciate your consideration. All corrections have been highlighted in red. The manuscript was once again revised to improve its English translation.
Reviewer 4 |
|
Suggestions/comments |
Response |
Dear authors, I have attached the comments to your work. Congratulations! Dear Authors, The manuscript ‘Understanding Medication Errors in Intensive Care Settings and Operating Rooms - A Systematic Review with Recommendations for Evidence-Based Practice’ is undoubtedly a contribution to updating knowledge in this specific area. In my opinion it can contribute to improve practical interventions in health services. The study demonstrates consistency throughout its sections, including the introduction, objectives, materials and methods, results, discussion, conclusions and implications for practice. It's just a suggestion, but because I think the title is too long, why not just : Medication Errors in Intensive Care Settings and Operating Rooms - A Systematic Review. Systematic literature reviews should provide implications for practice. They are essential for synthesizing existing evidence, identifying gaps in knowledge, and offering data-driven guidance for informed decision-making. It is suggested that acronyms should not be used for the first time without being preceded by the word written out in full with the acronym in brackets (ME Index and WHO), which can be seen in the abstract. The method has many positive aspects, including a careful selection of articles, clear definitions of inclusion criteria, and the use of tools for assessing study quality.
The text written in lines 116 and 117 appears in red, please correct it.
Applying clear criteria, such as a five-year limit for the publications included, is a good practice that helps to focus the review on the most current and relevant literature. It is suggested that the justification for this time limit, could be this (the most recent evidence, considering that no review was found carried out in this period). In the prisma flow chart, when we add up the articles from the different databases, it makes a total of 15 articles and not 13, which is the number mentioned by you for the full analysis. Please check The use of Joanna Briggs Institute tools to critically evaluate the methodological quality of included studies strengthens the review. Congratulations. Results are adequately described. Between lines 334 and 337 of the article, you state: “Five of the studies confirm that the introduction of new technologies at ICU’s including an electronic system of doctor’s recommendations, advance error reporting systems and pumps with a library of medications, can reduce the number of medication errors significantly”, but only 4 of the 5 studies are cited. Please confirm that. It presents a discussion where the results are confronted with other studies, as well as the healthcare impact. It also presents the implications and limitations of the study, as well as recommendations for practice. Conclusion is consistent with the objective. The references are adequate and in sufficient number. Congratulations on your work
|
Thank you for your valuable comment. We agree with your sugestion. The title has been changed into the Medication Errors in Intensive Care Settings and Operating Rooms - A Systematic Review.
Thank you for your valuable comment. We agree with your sugestion. We wrote the full explanation of the acronym. „…The review included 13 articles and original studies which met the PICOS-based criteria. The analyses confirmed that the operating theatre's medication error rate was 7.3% to 12%. In the case of Intensive Care Units, the medication error rate was from 1.32 to 31.7%. Conclusions: Medication errors in an operating room and intensive care is high. However, the values presented herein do not differ from the general Medication Error Index for medical centres, as calculated by the World Health Organization.
Thank you for your valuable comment. We agree with your suggestion. We corrected the sentence, and now it is in black colour.
Thank you for pointing this out. We agree with this comment. Therefore, we have add into the text justification for that limit. “…The most recent evidence showed that no review was carried out during this period, considering that type of inclusion criteria”.
Dear Reviewer, thank you for that comment. After evaluation of the Prisma chart, we correct the mistakes.
Thank you for your suggestion. We agree with your comment that the JBI tool is a very good tool for evaluating the review articles.
Thank you for pointing out that editing mistake. We correct it – „…Four articles analysed patient documentation, including the electronic history of orders [20,22,23,26].” We regret that we did not avoid such kind of errors
Thank you for all your comments and suggestions. We hope our work will improve patient safety and culture in the highly specialised units. It is a good recommendation for nursing managers, who greatly influence the hospital environment and the modification/ improvement of environmental factors. |

Round 2
Reviewer 2 Report
Comments and Suggestions for Authors
The manuscript has shown significant improvement, with many of the suggested revisions successfully incorporated and I appreciate the effort for revising the paper.